# Transforming Cardiotoxicity Detection in Cancer Therapies: The Promise of MicroRNAs as Precision Biomarkers

**DOI:** 10.3390/ijms252211910

**Published:** 2024-11-06

**Authors:** Isabel Moscoso, Moisés Rodríguez-Mañero, María Cebro-Márquez, Marta E. Vilar-Sánchez, Valentina Serrano-Cruz, Iria Vidal-Abeijón, María Amparo Martínez-Monzonís, Pilar Mazón-Ramos, Milagros Pedreira, José Ramón González-Juanatey, Ricardo Lage

**Affiliations:** 1Cardiology Group, Centre for Research in Molecular Medicine and Chronic Diseases (CIMUS), Instituto de Investigación Sanitaria de Santiago de Compostela (IDIS), University of Santiago de Compostela, 15782 Santiago de Compostela, Spain; maria.cebro.marquez@usc.es (M.C.-M.); marta.vilar.sanchez@rai.usc.es (M.E.V.-S.); valentina.serrano@rai.usc.gal (V.S.-C.); iria.vidal.abeijon@usc.es (I.V.-A.); 2Department of Cardiology and Coronary Unit, Instituto de Investigación Sanitaria de Santiago de Compostela (IDIS), Complexo Hospitalario Universitario de Santiago de Compostela, 15706 Santiago de Compostela, Spain; moises.rodriguez.manero@sergas.es (M.R.-M.); maria.amparo.martinez.monzonis@sergas.es (M.A.M.-M.); pilar.mazon.ramos@sergas.es (P.M.-R.); milagros.pedreira.perez@sergas.es (M.P.); jose.ramon.gonzalez.juanatey@sergas.es (J.R.G.-J.); 3Centro de Investigación Biomédica en Red de Enfermedades Cardiovasculares (CIBERCV), 28029 Madrid, Spain; 4Department of Biochemistry and Molecular Biology, Faculty of Medicine, Instituto de Investigación Sanitaria de Santiago de Compostela (IDIS), University of Santiago de Compostela, 15782 Santiago de Compostela, Spain

**Keywords:** cardiotoxicity, biomarkers, microRNAs

## Abstract

Cardiotoxicity (CDTX) is a critical side effect of many cancer therapies, leading to increased morbidity and mortality if not addressed. Early detection of CDTX is essential, and while echocardiographic measures like global longitudinal strain offer promise in identifying early myocardial dysfunction, the search for reliable biomarkers continues. MicroRNAs (miRNAs) are emerging as important non-coding RNA molecules that regulate gene expression post-transcriptionally, influencing key biological processes such as the cell cycle, apoptosis, and stress responses. In cardiovascular diseases, miRNAs have demonstrated potential as biomarkers due to their stability in circulation and specific expression patterns that reflect pathological changes. Certain miRNAs have been linked to CDTX and hold promise for early detection, prognosis, and therapeutic targeting. These miRNAs not only assist in identifying early cardiac injury, but also offer opportunities for personalized interventions by modulating their expression to influence disease progression. As research advances, integrating miRNA profiling with traditional diagnostic methods could enhance the management of CDTX in cancer patients, paving the way for improved patient outcomes and more tailored therapeutic strategies. Further clinical studies are essential to validate the clinical utility of miRNAs in managing CDTX.

## 1. Introduction

Cardiotoxicity (CDTX) is an important side effect of many cancer treatments, and can lead to long-term morbidity and mortality if not identified and managed promptly. Early detection of CDTX is crucial, and one promising method for this is the assessment of global longitudinal strain (GLS) by echocardiography. Unlike traditional measures such as left ventricular ejection fraction (LVEF), GLS can detect subtle changes in myocardial function before overt heart failure symptoms emerge, enabling earlier intervention [1]. In addition to imaging techniques, emerging biomarkers offer potential for early diagnosis and monitoring of CDTX, such as cardiac troponin or natriuretic peptides, but their use as biomarkers of CDTX has several limitations [2,3,4,5].

MicroRNAs (miRNAs) are small, non-coding RNA molecules, 19–25 nucleotides in length, that play a fundamental role in the post-transcriptional regulation of gene expression [6]. miRNAs biogenesis starts with the transcription of pri-miRNA by RNA polymerase II or III. This long transcript (~100 bp) is processed in the nucleus by the enzyme DROSHA into pre-miRNA (~70 bp). Pre-miRNAs are exported to the cytoplasm via exportin 5, where they are processed by the RNase enzyme DICER, which breaks the terminal loop, producing a mature RNA duplex (~22 bp). The less stable strand is loaded into Argonaute proteins (AGO2), forming the RNA-induced silencing complex (RISC). This complex enables the miRNA to bind to target mRNAs, leading to their degradation or inhibition of translation, depending on the degree of base pairing. A single miRNA can regulate thousands of mRNAs, and conversely, an mRNA can be targeted by many miRNAs (Figure 1) [7].

miRNAs function by binding to complementary sequences on target messenger RNAs (mRNAs), resulting in mRNA degradation or inhibition of translation [8]. This regulatory capacity sets miRNAs as crucial players in a high number of biological processes, including cell cycle control, differentiation, apoptosis, and stress response. In the context of cardiovascular disease (CVD), they have emerged as significant molecules of interest since miRNAs are related to different physiological and pathophysiological processes, and have also shown promise as CVD biomarkers [9]. Alterations in miRNA expression can reflect pathological conditions such as myocardial infarction, heart failure, and cardiomyopathy [10]. The pathogenesis of CVDs is often complex and multifactorial, involving genetic, environmental, and lifestyle factors. miRNAs are intimately involved in the development and progression of CVDs as part of intricate regulatory networks. The potential of miRNAs as biomarkers in CVD is due to several of their key characteristics. Firstly, miRNAs are remarkably stable in circulating blood and other body fluids, bearing severe conditions that typically degrade other nucleic acids, which makes them ideal candidates for non-invasive biomarkers. This stability is partially due to their encapsulation within exosomes, microvesicles, or binding to proteins, which protects them from RNase activity [8]. This stability, combined with their specificity to different stages and types of heart disease, enhances their potential utility in clinical settings. Secondly, the expression profiles of specific miRNAs are altered in response to pathological changes in cardiovascular tissues, reflecting the underlying disease state [8]. The diagnostic and prognostic capabilities of miRNAs are complemented by their potential therapeutic applications [11]. Moreover, miRNAs offer prognostic value. Certain miRNAs have been linked to adverse cardiovascular events and their levels can predict disease outcomes, thereby improving risk stratification. miRNAs are emerging as multifaceted tools in cardiovascular area, offering promise not only as biomarkers for early detection and prognosis of CDTX, but also as potential therapeutic targets. Therapeutically, miRNAs hold promise as targets for novel interventions. Modulating the expression of specific miRNAs could potentially alter disease progression, offering a new avenue for treatment. miRNAs represent a successful research area in patients in vitro or in vivo, with significant implications for cardiovascular medicine (Figure 2) [12].

miRNAs’ role as biomarkers offers the potential for early and accurate diagnosis, effective monitoring of disease progression, and personalized therapeutic strategies. As research advances, integrating miRNA profiling into clinical practice could transform the management of CVDs, improving patient outcomes and reducing the global burden of these conditions [13]. Further research into the specific roles and mechanisms of miRNAs in heart disease will likely unlock new avenues for managing and treating CVDs [9]. The integration of miRNA profiling with traditional and novel imaging techniques could enhance the predictive power for early CDTX, allowing for more personalized and effective management strategies in cancer patients [2,14].

This review focuses on the potential role of miRNAs as biomarkers for CDTX. First, it explores specific miRNAs implicated in cardiovascular health and disease, their mechanisms of action, and some of their potential clinical applications. By understanding the involvement of miRNAs in CDTX, we can better identify and manage this significant side effect of cancer treatments.

## 2. MiRNAs Involved in CDTX

### 2.1. Let-7

The Let-7 family plays crucial roles in various physiological and pathological processes, including CVDs. As potential biomarkers for CDTX, members of this family, such as Let-7a, Let-7b, Let-7c, and others, exert their influence on cardiac health through specific mechanisms. The Let-7 family is involved in the regulation of apoptosis, a key process in CDTX. Let-7 miRNAs can target and downregulate anti-apoptotic genes like *Bcl-xl* and *Bcl-2* [15], promoting apoptosis in cardiomyocytes under stress and contributing to CDTX. The Let-7 miRNAs also play a role in regulating cardiac hypertrophy, a condition also linked to CDTX. Members of the Let-7 family can affect hypertrophic signaling by targeting genes involved in pathways such as the *PI3K/Akt* and *mTOR*. For instance, Let-7 targets the signaling molecule CALM [16], which is also implicated in hypertrophy. Let-7 miRNAs regulate cardiac cell growth, preventing pathological hypertrophy that can lead to heart failure and exacerbate CDTX. Furthermore, the Let-7 family can increase CDTX modulating fibrosis-related pathways by targeting genes such as collagen genes, affecting fibrotic remodeling in the heart [17]. Let-7 miRNAs inhibit excessive fibrotic remodeling, thereby preserving cardiac function that could mitigate CDTX. Let-7 miRNAs are also involved in modulating inflammatory responses, which are critical in CDTX. This family of miRNAs targets inflammatory cytokines and signaling molecules such as *TLR4*, impacting the inflammatory response in cardiac tissues. Let-7 miRNAs reduce cardiac inflammation, thus mitigating cardiac injury associated with CDTX [15]. The Let-7 family contributes to the cellular response to oxidative stress, a significant factor in CDTX. Let-7 miRNAs can target genes involved in the oxidative stress response, such as *HMOX1* [18]. Let-7 miRNAs reduce oxidative stress and damage to cardiac tissues, thus alleviating the effects of CDTX. Additionally, Let-7 miRNAs are involved in cardiac development and repair processes. This family can influence the expression of genes related to cardiac development and repair, affecting the heart’s regenerative capacity. Regulating these developmental pathways, Let-7 miRNAs impact the repair processes following cardiac injury, which influences the overall outcome of CDTX [19]. In vitro and in vivo experiments with doxorubicin (DOX)-treated rats, isolated neonatal rat cardiomyocytes, and H9c2 cardiomyocytes have shown a decrease in Let-7 expression [20]. In porcine models of CDTX induced by DOX treatment, Let-7 expression was found to be downregulated [21]. Deep sequencing analysis also revealed differential expression of the Let-7 family in H9c2 cardiomyocytes exposed to DOX [22]. Additionally, experiments with rats and primary cultured myocardial cells treated with DOX indicate that Let-7g is implicated in myocardial injury [23]. Also, Let-7f was found to be significantly downregulated in anthracyclines treated patients suffering from CDTX [24].

### 2.2. miR-1

miR-1 is predominantly expressed in the cardiac and skeletal muscles and plays a crucial role in regulating cardiac development, differentiation, and conduction. Elevated miR-1 levels have been linked to myocardial infarction, heart failure, and arrhythmias [25]. As a muscle-specific miRNA, miR-1 has been extensively considered for its significance in cardiac biology and its potential as a biomarker for cardiac dysfunction. miR-1 is essential for regulating cardiac ion channels, which are vital for maintaining proper cardiac electrophysiology. Dysregulation of these channels can lead to arrhythmias and other forms of CDTX. Specifically, miR-1 negatively regulates ion channel genes such as *KCNJ2* and *GJA1*; changes in these genes can affect cardiac action potential duration and conduction velocity, potentially leading to arrhythmias [26]. Additionally, miR-1 is involved in the regulation of cardiac hypertrophy and apoptosis, both critical processes in the development of CDTX, suppressing hypertrophic signaling pathways by targeting genes like *IGF-1* and its receptor, *IGF1R* [27]. Increased miR-1 levels are associated with cardiac dysfunction, potentially due to its role in promoting oxidative stress and cardiomyocyte apoptosis, mechanisms linked to DOX-induced CDTX [28]. miR-1 can promote apoptosis in cardiomyocytes, rat heart tissue, and H9c2 by downregulating anti-apoptotic genes such as *Bcl-2* and *HSP60* [29,30]. Furthermore, miR-1 responds to oxidative stress conditions by targeting genes encoding antioxidant enzymes and increasing oxidative stress through downregulation of SOD1 expression, which is crucial for detoxifying superoxide radicals [31]. Research has shown that circulating miR-1 levels correlate with changes in LVEF and GLS in breast cancer patients undergoing DOX treatment, providing a better predictive value than cardiac troponin I (cTNI) [20].

### 2.3. miR-106b-5p

miR-106b-5p has been explored for its involvement in CVDs and its potential as a biomarker for CDTX. This miRNA plays a key role in regulating apoptosis. Some studies show that miR-106b-5p targets and downregulates pro-apoptotic genes such as *FASTK*, *BCL11B*, and *CDKN1A*, leading to reduced apoptosis and enhanced cell survival in stressed H9c2 animal models and the human heart after myocardial infarction [32]. Following these results, miR-106b-5p could support cardiomyocyte survival and mitigate the apoptotic cell death associated with CDTX. However, miR-106b-5p was reported to increase apoptosis in IPS-CM cells after DOX treatment via Casp3/7 [33]. miR-106b-5p also impacts the regulation of the cell cycle, influencing cardiac cell proliferation and repair. It has been reported that miR-106-3p targets cell cycle inhibitors like p21 and p27, which are essential for cell cycle arrest. Downregulating these inhibitors promotes cell cycle progression and cell proliferation, increasing the repair of damaged cancer cells [34]; these mechanisms could influence and reduce the effects of CDTX. Additionally, miR-106b-5p is involved in the regulation of angiogenesis, which is crucial for maintaining cardiac function. It targets and upregulates factors such as *FAT4* and *VEGF*, promoting new blood vessel formation and improving blood supply to cardiac tissues. miR-106b-5p supports better vascularization of the heart, which alleviates cardiotoxic effects [35]. miR-106b-5p also regulates cardiac fibrosis, which can worsen CDTX. It modulates the expression of *TGF-β* and collagen genes, influencing the extent of fibrotic remodeling in the heart and preventing pathological fibrotic remodeling, as well as preserving cardiac function and reducing CDTX impact [36]. In response to oxidative stress, miR-106b-5p enhances the antioxidant capacity of neonatal mouse cardiomyocytes. Through its regulatory actions, miR-106b-5p reduces oxidative damage to cardiac tissues, thus reducing the effects of CDTX [37]. miR-106b-5p has also been shown to modulate inflammatory pathways critical in CDTX, regulating the expression of inflammatory cytokines and signaling molecules such as IL-6 and TNF-α. Regulating these molecules, miR-106b-5p impacts the inflammatory response in the heart, helping to decrease cardiac injury and slow CDTX progression [38]. On the other hand, miR-106a promotes cardiac hypertrophy by targeting *MFN-2* [39]. Similarly, silencing miR-106b-5p has been found to prevent DOX-induced CDTX by modulating the PR55α/YY1/sST2 signaling axis in both in vitro iPS-CM and in vivo mice models [33].

### 2.4. miR-126

miR-126 is an endothelial-specific miRNA crucial for vascular integrity and angiogenesis, making it significant in cardiovascular health and disease. Its potential as a biomarker for CDTX involves several specific mechanisms. miR-126 is a key regulator of angiogenesis, essential for maintaining healthy cardiac tissue and function. It enhances angiogenesis by targeting inhibitors of the VEGF signaling pathway, such as *SPRED1* and *PIK3R2*, promoting endothelial cell proliferation and new blood vessel formation. Facilitating angiogenesis, miR-126 ensures an adequate blood supply to cardiac tissues, and is crucial for preventing and mitigating ischemic injury often seen in CDTX [40]. miR-126 also plays a vital role in maintaining vascular integrity and proper endothelial function by targeting genes as *VCAM-1*; modulating these genes, miR-126 maintains endothelial barrier function and reduces vascular inflammation, protecting against endothelial dysfunction and vascular injury [41]. Additionally, miR-126 has anti-inflammatory properties that contribute to its role in CDTX by targeting multiple inflammatory signaling pathways, including those involving NF-κB/PI3K/AKT/mTOR [42]. miR-126 reduces inflammatory responses in the cardiovascular system, lowering the production of pro-inflammatory cytokines and thereby reducing inflammation-induced damage in cardiac tissues [43]. Oxidative stress is another condition to which miR-126 responds, regulating the activity of proteins involved in the oxidative stress response, such as SOD and GPx, and alleviating the damaging effects of reactive oxygen species (ROS) on endothelial cells. Through its regulatory actions, miR-126 protects cardiac tissues from oxidative damage, ameliorating CDTX effects [44]. Moreover, miR-126-3p protects human cardiac endothelial cells from hypoxia/reoxygenation-induced injury and inflammatory response by activating the PI3K/Akt/eNOS signaling pathway. In addition, increased miR-126-3p expression has been reported in hiPSC-CM in vitro after DOX treatment [45] and in breast cancer patients after neoadjuvant treatment [46]. However, miR-126 has been shown to be significantly downregulated in different studies of breast cancer patients experiencing cardiotoxic effects after neoadjuvant treatment [24,47,48].

### 2.5. miR-129

miR-129-5p has been studied for its involvement in various diseases, including CVDs, and as a potential biomarker for CDTX. This miRNA plays a critical role in regulating apoptosis; miR-129-5p inhibits autophagy and apoptosis in cardiomyocytes through the PI3K/AKT/mTOR signaling pathway by targeting ATG14 [49]. miR-129-5p is also implicated in the regulation of cardiac hypertrophy by targeting genes involved in hypertrophic signaling pathways, such as *KEAP1*. Through its influence on hypertrophic signaling, miR-129-5p prevents pathological cardiac hypertrophy, which is often linked to cardiac dysfunction and CDTX [50]. Furthermore, miR-129-5p is involved in regulating cardiac fibrosis, modulating the expression of fibrosis-related genes, such as *COL1A1* and *TGF-β*. The regulation of this mechanism impacts the extent of fibrosis in the heart. miR-129-5p inhibits pathological fibrotic remodeling, thus preserving cardiac function and reducing the impact of CDTX [51]. In addition, miR-129-5p has been shown to modulate inflammatory pathways, having an influence on the expression of inflammatory cytokines and signaling molecules, such as IL-6 and TNF-α. miR-129-5p impacts the inflammatory response, helping to reduce cardiac injury and the progression of CDTX [52]. miR-129-5p also plays a role in the cellular response to oxidative stress, targeting genes such as *Nrf2* and *SOD2*, which are involved in the oxidative stress response. Thus, miR-129-5p enhances the antioxidant capacity of human cardiomyocytes, reducing oxidative damage to cardiac tissues [50]. Moreover, miR-129-5p may directly target *HMGB1*, whose overexpression worsens hypoxia-induced myocardial injury. Regulating the miR-129-5p/*HMGB1* axis reduces TKI-induced injury in H9c2 cells [53]. Additionally, increasing miR-129-1-3p expression in HL1 cells in vitro has been shown to reduce anthracycline-induced CDTX [54].

### 2.6. miR-133

miR-133, together with miR-1, is a muscle-specific miRNA crucial for cardiac function and a potential biomarker for CDTX. The specific mechanisms by which miR-133 can be used as a biomarker for CDTX include its involvement in regulating cardiac hypertrophy and apoptosis. miR-133 negatively regulates hypertrophic signaling pathways by targeting key genes like *RhoA*, *Cdc42*, and *Nelf-A/WHSC2* [55], preventing pathological cardiac growth, a hallmark of CDTX. Additionally, miR-133 suppresses apoptosis in rat cardiomyocytes by targeting pro-apoptotic genes such as *Casp-9* and *Apaf-1*, protecting the heart from injury-induced cell death [56]. Similar to miR-1, miR-133 is involved in regulating cardiac ion channels, which is essential for maintaining proper cardiac electrophysiology. miR-133 regulates the expression of genes encoding ion channels, including *KCNJ2* and *CACNA1C*, and by modulating these channels, miR-133 maintains cardiac rhythm and function. Dysregulation of ion channels can lead to arrhythmias and other forms of CDTX, making miR-133 key to preserving cardiac electrophysiological stability [57]. Oxidative stress significantly contributes to CDTX, and miR-133 responds to changes in oxidative stress levels. miR-133 targets genes involved in the antioxidant defense system, such as *BACH1*, thereby protecting H9c2 from oxidative damage [31]. Through its regulatory actions, miR-133 could also reduce oxidative stress in cardiac tissues, mitigating CDTX. miR-133a also targets ErbB2/Her2/neu, a crucial factor in preventing anthracycline toxicity in cardiomyocytes. Loss of ErbB2 signaling in cardiomyocytes aggravates DOX-induced cell death, so inhibiting its activity could affect DOX-induced CDTX mechanisms [58]. In vivo assays using a mouse model of DOX-induced CDTX showed that miR-133b expression was significantly downregulated. Overexpressing miR-133b increased Bcl-2 protein expression and decreased Bax and cleaved caspase-3 protein expression. miR-133b binds to the 3′-UTR of *PTBP1* and *TAGLN2* mRNAs to regulate downstream targets, suggesting that its protective effect on cardiomyocyte apoptosis and myocardial fibrosis may occur through inhibiting *PTBP1* and *TAGLN2* expression [20].

### 2.7. miR-140

The miR-140 family, comprising miR-140-3p and miR-140-5p, plays roles in various physiological and pathological processes, including CVDs. miR-140 is involved in regulating apoptosis, targeting and downregulating pro-apoptotic genes associated with the mitochondrial pathway of apoptosis, such as *YES1*. miR-140-3p promotes cell survival in cardiomyocytes under stress inhibiting apoptosis. Similarly, miR-140-5p influences apoptosis by targeting genes related to cell death pathways, balancing the apoptotic response in mouse hearts, which is crucial for CDTX [59]. miR-140 impacts cardiac hypertrophy, modulating hypertrophic signaling by targeting genes like *PI3K/Akt* and *mTOR*. This modulation affects cardiac cell growth and hypertrophy, helping to prevent or relieve pathological cardiac hypertrophy. Also, the regulation of cardiac fibrosis is also influenced by miR-140, which alters the expression of fibrosis-related genes such as *TGF-β* and *Smad3*, impacting the extent of fibrosis in the heart and preventing excessive fibrotic remodeling, thereby preserving cardiac function and reducing CDTX impact [60]. miR-140 has been shown to modulate inflammatory pathways critical in CDTX, modulating inflammatory cytokines and signaling molecules, including IL-6 and TNF-α, in other pathologies. This regulation could also impact the inflammatory response in cardiac tissues, reducing cardiac inflammation and mitigating cardiac injury and CDTX progression [61]. In response to oxidative stress, miR-140-3p targets genes such as *Nrf2* and *Sirt2* [62,63]. In vivo murine models of DOX-induced CDTX showed an upregulation of miR-140 expression. Additionally, in in vitro experiments with DOX-treated H9c2 cells and cardiac fibroblast models, miR-140-5p expression levels were significantly increased [20,64]. Moreover, miR-140-5p mediates bevacizumab-induced cytotoxicity in human cardiomyocytes by targeting the VEGFA/14-3-3γ signaling pathway [65].

### 2.8. miR-143

miR-143 has been identified as playing significant roles in various CVDs and is under investigation as a potential biomarker for CDTX. This miRNA can promote apoptosis by targeting and downregulating anti-apoptotic genes such as *Bcl-2*, which increases the susceptibility of rat neonatal cardiomyocytes to programmed cell death [66]. This miRNA targets genes involved in the differentiation and proliferation of vascular smooth muscle cells, such as *ELK1* and *PDGF*, which maintain vascular integrity and function [67]. Additionally, miR-143 modulates genes affecting endothelial cell function, such as *eNOS*, that could have an impact on vasodilation and vascular health, crucial factors in CDTX [68]. Furthermore, miR-143 is involved in regulating fibrosis and modulating the expression of fibrosis-related genes, including *SPRY3* and collagen genes [69]. This regulation influences the extent of fibrosis, helping to increase pathological fibrotic remodeling, thereby reducing cardiac function and intensifying CDTX effects. miR-143 also affects metabolic pathways related to cardiac function, targeting genes involved in lipid metabolism, such as *AKT* [70] and *PPARγ* [71], and influencing lipid accumulation and the metabolic health of cardiomyocytes. Additionally, miR-143 impacts glucose metabolism by targeting genes like *GLUT4*, affecting energy availability and utilization in mouse preadipocytes under stress, cells closely related to cardiovascular health [71]. In the context of oxidative stress, miR-143 targets genes involved in the oxidative stress response, such as *ATG7* [72]. By modulating these genes, miR-143 enhances the antioxidant capacity of mouse cardiac progenitor cells, reducing oxidative damage to cardiac tissues and mitigating CDTX effects. miR-143 has also been shown to modulate inflammatory pathways, influencing the expression of inflammatory cytokines and signaling molecules and impacting the inflammatory response in the heart [73]. Reducing inflammation, miR-143 mitigates cardiac injury and slows CDTX progression. Bioinformatic analyses and experimental studies in DOX-induced CDTX models, both in vitro and in vivo, indicate that miR-143-5p expression is decreased and affects oxidative stress and apoptosis by targeting the *HIF-1* and *PI3K-Akt* signaling pathways [74] or inhibiting *AKT* [75]. Conversely, miR-143 is found to be downregulated in DOX-induced CDTX in H9c2 cells and murine models [76].

### 2.9. miR-194-5p

miR-194-5p is increasingly recognized for its roles in cardiovascular health and disease, including CDTX. It is involved in regulating apoptosis by targeting genes that are part of apoptotic signaling pathways. For instance, it can impact the expression of pro-apoptotic genes such as *Caspase-3*, *Bax*, and *Bcl-2* [77], affecting cardiomyocyte survival under stress, which is particularly relevant in CDTX, where excessive cell death can lead to cardiac dysfunction. Additionally, miR-194-5p plays a role in the cellular response to oxidative stress by targeting genes such as *SOD* and *MDA* [77]. This regulation influences the antioxidant capacity of H9c2 cells and reduces oxidative damage to cardiac tissues, offering benefits in CDTX management. miR-194-5p may also impact cardiac hypertrophy associated with CDTX. It can target genes involved in hypertrophic signaling pathways, including those in the CnA/NFATc2 pathway [78]. By modulating these pathways, miR-194-5p can help to prevent or reduce excessive cardiac hypertrophy, a common feature in CDTX. Furthermore, it has been shown, both in vitro and in vivo, that miR-194-5p reduces DOX-induced cardiomyocyte apoptosis and endoplasmic reticulum stress by targeting *PAK2* and *XBP1s* in H9c2 cells and murine models [79].

### 2.10. miR-199

miR-199a-5p has been investigated for its role in CVDs and its potential as a biomarker for CDTX. This miRNA is involved in various cellular processes and pathways that impact cardiac function and response to injury. miR-199a-5p is crucial in regulating apoptosis. This miRNA targets and downregulates pro-apoptotic genes such as *HIF-1α* and Caspase-3, and by reducing their expression, miR-199a-5p prevents programmed cell death in rat cardiomyocytes [80]. Through its anti-apoptotic effects, miR-199a-5p promotes cardiomyocyte survival under stress, which is essential for mitigating CDTX. miR-199a-5p also plays a role in regulating cardiac hypertrophy by targeting genes like *PGC1α*, which are crucial for cardiac cell growth and hypertrophy [81]. Modulating these pathways, miR-199a-5p prevents pathological cardiac hypertrophy, thereby reducing the risk of heart failure and other forms of CDTX. Furthermore, miR-199a-5p is involved in regulating autophagy, a cellular process vital for maintaining cardiac homeostasis. It targets and downregulates genes such as *Beclin-1* and *ATG7* [82], and this modulation affects the balance of autophagy in cardiomyocytes. Proper autophagy regulation is critical for cardiomyocyte survival, especially under stress. Dysregulation of autophagy by miR-199a-5p can lead to increased cell death and CDTX. In the context of oxidative stress, miR-199a-5p targets and influences the expression of antioxidant genes such as *SIRT1*, *SOD1*, and, indirectly, *eNOS* [83]. Regulating these genes, miR-199a-5p enhances the antioxidant capacity of human endothelial cells, helping to reduce oxidative damage to cardiac tissues and mitigate the effects of CDTX. miR-199a-5p is also involved in regulating cardiac fibrosis, affecting the expression of fibrosis-related genes, such as *TGF-β* and collagen genes [84]. miR-199a-5p inhibits pathological fibrotic remodeling, preserving cardiac function and decreasing the impact of CDTX. Additionally, downregulation of miR-199a-5p increases fluorouracil-induced CDTX in vitro by activating the *ATF6* signaling pathway in primary cardiomyocytes from neonatal rats [85]. Moreover, miR-199-3p was found to be elevated in breast cancer patients receiving neoadjuvant therapy [86].

### 2.11. miR-200a

miR-200a, a member of the miR-200 family, has been studied for its involvement in various physiological and pathological processes, including its role in CVDs and its potential as a biomarker for CDTX. miR-200a plays a significant role in regulating apoptosis by targeting anti-apoptotic genes such as *Bcl-2* and *Bax* [87], increasing cardiac cell susceptibility to apoptosis. By enhancing the expression of pro-apoptotic genes and inhibiting anti-apoptotic ones, miR-200a contributes to cardiomyocyte death, leading to cardiac injury and dysfunction. miR-200a is also involved in regulating oxidative stress and targeting genes such as *SOD2* and *Nrf2*; modulating these genes affects the cellular antioxidant capacity. Reduced expression of protective antioxidant genes increases oxidative damage to cardiac tissues, worsening cardiotoxic effects [88]. Additionally, miR-200a regulates fibrotic processes in the heart, modulating the expression of *TGF-β* and *ZEB1* and influencing fibrotic remodeling. Elevated levels of miR-200a can enhance fibrotic responses, leading to increased extracellular matrix protein deposition and stiffening of cardiac tissue [89]. miR-200a has also been shown to modulate inflammatory pathways, targeting and modulating the expression of cytokines and signaling molecules like TNF-α and IL-6, potentially increasing inflammatory responses. Promoting inflammation, miR-200a exacerbates cardiac injury and contributes to CDTX progression [90]. In an in vivo mouse model of DOX-induced CDTX, miR-200a expression was decreased, affecting ferroptosis by activating the Nrf2 signaling pathway [91]. This suggests that miR-200a decreases DOX-induced CDTX through upregulation of *Nrf2* in mice [92]. In contrast, miR-200a-3p increases DOX-induced CDTX by targeting *PEG3* through the SIRT1/NF-κB signaling pathway in an in vivo murine model [93].

### 2.12. miR-208

miR-208, a cardiac-specific miRNA, plays a significant role in heart disease and holds potential as a biomarker for CDTX. miR-208 is crucially involved in cardiac hypertrophy, regulating the expression of *MYH7* and *MYH6*, which are essential for cardiac muscle function and structure. miR-208 promotes *MYH7* expression while repressing *MYH6*, facilitating the hypertrophic response, contributing to cardiac remodeling [94]. miR-208 also plays a role in the cardiac stress response, regulating the expression of the *THRAP1* and *MED13* genes. These genes adapt the heart to various stressors, including mechanical stress and toxic insults. Regulating these stress-response pathways, miR-208 reduces the effects of cardiotoxic agents [95]. Cardiac fibrosis is another process in which miR-208 is involved; together with miR-21, miR-208 regulates the expression of *TGF-β1* and *Smad-3*. Increased expression of miR-208 is associated with enhanced fibrotic remodeling, compromising cardiac function and contributing to CDTX [96]. Furthermore, miR-208 affects cardiac contractility, an essential aspect of heart function that can be impaired in CDTX. miR-208 targets genes encoding proteins such as *MYH7*. Alterations in the expression levels can lead to changes in cardiac contractility, potentially contributing to the development of CDTX [97]. In different in vivo murine models, DOX-induced CDTX shows a decrease in miR-208a or an increase in miR-208b [20]. Additionally, in other CDTX models, rats exposed to isoproterenol exhibit increased plasma levels of miR-208, along with elevated cardiac troponin levels [98].

### 2.13. miR-21

miR-21 is a potential biomarker for cardiac remodeling and fibrosis, with its inhibition potentially offering therapeutic benefits. As one of the most studied miRNAs in CVD, miR-21 has emerged as a significant biomarker for CDTX. It plays a critical role in cardiac fibrosis by targeting several key antifibrotic genes such as *SPRY1* and *PDCD4*, enhancing fibroblast proliferation and collagen production, upregulating the expression of MMPs involved in ECM remodeling, and contributing to the fibrotic response in the heart [99]. miR-21 also exerts anti-apoptotic effects through targeting pro-apoptotic genes like *PTEN* and *PDCD4*, which leads to increased cell survival, particularly in cardiac fibroblasts [20]. While miR-21 promotes survival in cardiac fibroblasts, its role in cardiac myocytes is complex and context-dependent, sometimes promoting survival and at other times leading to adverse remodeling [99]. Additionally, miR-21 responds to oxidative stress conditions by targeting genes involved in oxidative response, such as *SOD2* and *KRIT1* [31]. miR-21 manages oxidative stress levels in endothelial cells, indirectly affecting the production and accumulation of ROS, which are harmful to these cells. miR-21 also regulates the inflammatory response by targeting multiple signaling pathways, including *NF-κB* and *TGF-β* signaling. These pathways are crucial in the development of inflammation and fibrosis in the heart. miR-21 can enhance the production of pro-inflammatory cytokines, contributing to the inflammatory milieu that exacerbates cardiac injury [100]. In a mouse model of DOX-induced CDTX, increasing nucleolin/miR-21 levels prevented cardiomyocyte apoptosis, nucleolar stress, hypertrophy, and oxidative stress markers in the heart by knocking down *RGS6* expression [101]. In this sense, together with miR-1 and miR-29, miR-21 is increased in in vivo experiments involving rats treated with other cardiotoxic medications, such as antidepressants, which also increase oxidative stress [102]. Other in vivo murine models have shown that miR-21 protects cardiomyocytes against DOX-induced apoptosis by targeting *BTG2* [20,103]. In breast cancer patients treated with DOX, a decrease in circulating miR-21 expression has been observed [104].

### 2.14. miR-210

miR-210 is a hypoxia-inducible miRNA with a significant role in CVDs and has emerged as a potential biomarker for CDTX. It is highly induced under hypoxic conditions, which are often linked to cardiac injury and CDTX. miR-210 targets several hypoxia-related genes, including *ISCU* and *GPD1L*. miR-210 assists cells in adapting to low oxygen levels, which is crucial for maintaining cardiac function under stress. Under hypoxia, miR-210 promotes cell survival by inhibiting apoptosis and improving cellular adaptation to stress [105,106]. It is involved in the regulation of apoptosis, exerting anti-apoptotic effects by targeting pro-apoptotic genes such as *CASP8AP2*. This regulation decreases cell death in human cardiomyocytes and H9c2 cells, aiding in the preservation of cardiac tissue during injury. Modulating apoptosis, miR-210 enhances the survival of cardiomyocytes under stress and injury [105,106]. miR-210 also influences angiogenesis, crucial for maintaining cardiac tissue health and function. miR-210 targets and downregulates inhibitors of angiogenesis, like *EFNA3* and *PTP1B* in cancer [107]; this downregulation could promote the formation of new blood vessels. This mechanism is necessary for supplying oxygen and nutrients to cardiac tissues. Improved angiogenesis enhances blood flow to the heart, potentially mitigating the effects of ischemia and other forms of cardiac stress. miR-210 is involved in regulating mitochondrial function, which is essential for energy production in the heart. It targets genes related to mitochondrial metabolism and function, such as *ISCU* and *SDHD* also in cancer cells [107]. miR-210 supports mitochondrial function under stress, ensuring adequate energy production during hypoxic conditions, which is critical for maintaining cardiac function. miR-210 also responds to oxidative stress, influencing the expression of involved genes, such as *COX10* in cancer cells [107]. By boosting antioxidant defenses, miR-210 can protect cardiomyocytes from oxidative damage, contributing to overall cardiac protection. Along with other miRNAs, miR-210 is notably downregulated in breast cancer patients experiencing cardiotoxic effects following neoadjuvant treatment [24,47,48].

### 2.15. miR-34a-5p

miR-34a-5p is a miRNA that has attracted attention for its role in CVDs, including its potential as a biomarker for CDTX. Known for its strong pro-apoptotic effects, miR-34a-5p plays a significant role in CDTX. It targets and downregulates anti-apoptotic genes such as *Bcl-2* and *SIRT1*, promoting apoptosis in H9c2 cells and a rat CDTX-model. Thus, miR-34a-5p raises the rate of cardiomyocyte death, leading to cardiac injury and dysfunction [108]. miR-34a-5p is also involved in regulating fibrotic processes, modifying the expression of *TGF-β* and collagen genes, which leads to increased deposition of extracellular matrix proteins and fibrotic remodeling. Elevated expression levels of miR-34a-5p can enhance fibrotic responses, resulting in stiffening and dysfunction of cardiac tissue [109]. In addition, miR-34a-5p modulates inflammatory pathways modulating the expression of cytokines and signaling molecules, including IL-6 and TNF-α [110]. This can increase inflammatory responses, exacerbating cardiac injury and advancing CDTX. miR-34a-5p is also involved in regulating cardiac hypertrophy by targeting genes such as *MYC* and cyclins [109], which are involved in cell growth and proliferation. Increased expression of miR-34a-5p can lead to maladaptive cardiac hypertrophy, contributing to pathological heart remodeling. Furthermore, miR-34a-5p plays a role in the cellular response to oxidative stress, targeting *SIRT1* and *eNOS* genes [110]. Downregulation of these genes reduces the antioxidant capacity of human smooth muscle cells, downregulating the expression of protective antioxidant genes. miR-34a-5p increases oxidative damage to cardiac tissues, worsening cardiotoxic effects. In vitro and animal model studies have demonstrated that inhibiting miR-34a-5p regulates DOX-induced autophagy and inhibits myocardial pyroptosis via the Sirt3-AMPK pathway [111]. Both in vitro and in vivo studies reveal an upregulation of miR-34a-5p expression following DOX-induced CDTX. Piegari et al. showed that miR-34a is highly expressed in the plasma of rats treated with DOX and, although anthracycline administration upregulated miR-34a in the liver, kidney, and skeletal muscle, heart expression is outstanding [112]. Additionally, studies in breast cancer patients and mice models have shown that miR-34a-5p is also upregulated in DOX-induced CDTX [20,86,113].

### 2.16. miR-4732-3p

miR-4732-3p, a member of the miR-4732 family, which comprises various miRNAs involved in regulating cardiovascular function and stress responses, has shown potential as a biomarker for CDTX. Although research on miR-4732-3p is less extensive than for some other miRNAs, it plays a role in apoptosis regulation. miR-4732-3p from extracellular vesicles derived from human mesenchymal stromal cells targets pro-apoptotic genes or pathways, such as those involving caspases [114]. Inhibiting these genes, miR-4732-3p may reduce cardiomyocyte apoptosis and influence cell survival under stress, potentially mitigating CDTX effects. miR-4732-3p might also impact cardiac hypertrophy by targeting genes such as *PI3K/Akt* and *mTOR*. This modulation can affect the hypertrophic response in rat heart injury models and help prevent excessive cardiac hypertrophy. Additionally, miR-4732-3p may affect cardiac fibrosis, influencing the expression of collagen genes and impacting the fibrotic response in the heart [114]. miR-4732-3p may help to decrease pathological fibrotic remodeling, thereby preserving cardiac function and reducing CDTX severity. miR-4732-3p is also involved in the cellular response to oxidative stress; this miRNA can target genes related to oxidative stress responses, influencing the antioxidant capacity of neonatal rat cardiomyocytes [114]. Furthermore, miR-4732-3p may impact cardiac development and repair processes targeting genes involved in cell proliferation and differentiation; this regulation affects the ability of the heart to recover from injury [114]. Modulating these developmental pathways, miR-4732-3p can enhance cardiac repair mechanisms, which is crucial for reducing CDTX effects. It has been shown that miR-4732-3p is downregulated in breast cancer patients with CDTX and in hiPSC-CMs in vitro. Its therapeutic delivery has been shown to protect the heart from DOX-induced oxidative stress in rats [115,116].

### 2.17. miR-494-3p

miR-494-3p has been studied for its involvement in various CVDs, with growing interest in its potential as a biomarker for CDTX. This miRNA plays a crucial role in regulating apoptosis. miR-494-3p targets and downregulates pro-apoptotic genes such as *PTEN*, which results in reduced apoptosis and promotes cell survival in cardiomyocyte cells under stress [117]. Enhancing anti-apoptotic factors and inhibiting pro-apoptotic ones, miR-494-3p improves cardiomyocyte survival, potentially alleviating CDTX effects. In addition, miR-494-3p is involved in regulating cardiac fibrosis, which can worsen CDTX, inducing fibrosis in cardiac fibroblasts by inhibiting *PTEN* and boosting the phosphorylation of AKT, SMAD2/3, and ERK [118]. Through these pathways, miR-494-3p prevents pathological fibrotic remodeling, preserving cardiac function and reducing CDTX impact. miR-494-3p has also been shown to modulate inflammatory pathways critical in CDTX; it can influence the expression of IL-6 and TNF-α [117]. miR-494-3p impacts the inflammatory response in the heart, helping to reduce inflammation, mitigate cardiac injury, and slow CDTX progression. Moreover, miR-494-3p plays a role in the cellular response to oxidative stress, targeting genes including *NF-κB* [119]. miR-494-3p enhances the antioxidant capacity of rat neonatal cardiomyocytes modulating these genes, reduces oxidative damage to cardiac tissues, and alleviates CDTX effects. miR-494-3p is also known to promote angiogenesis, which can be protective in the context of CDTX. It targets and upregulates pro-angiogenic factors such as VEGF and CD31 [120]. Enhancing angiogenesis, miR-494-3p improves blood supply to the heart and supports cardiac function under stress. This increase in vascularization of cardiac tissue is beneficial for mitigating CDTX effects. In vitro assays in H9c2 and HL1 cells have shown that miR-494-3p expression mediates cell viability, oxidative damage, and apoptosis induced by anthracycline-induced CDTX [121,122].

### 2.18. miR-499

miR-499, a cardiac-specific miRNA, has garnered significant interest as a biomarker for CDTX due to its involvement in various cardiac processes. Both in vivo and in vitro experiments have shown miR-499-5p downregulation after DOX exposure [20]. miR-499 is deeply involved in the regulation of apoptosis by targeting pro-apoptotic genes such as *PDCD4* and *SOX6*, promoting cell survival, and protecting neonatal rat ventricular cardiomyocytes from apoptosis [123]. miR-499 also plays a role in regulating cardiac hypertrophy, inhibiting the expression of the *MYH7B*, *KCNH6*, and *ACTA1* genes. miR-499 prevents pathological cardiac hypertrophy, maintaining a balance between hypertrophic and anti-hypertrophic signals and thereby protecting the heart from excessive growth that can lead to dysfunction [29]. In plasma samples from patients undergoing coronary artery bypass grafts, miR-499 responds to oxidative stress conditions by increasing the expression of antioxidant genes and proteins, helping to reduce the damaging effects of ROS in human hearts. Through its regulatory actions, miR-499 reduces oxidative damage to the human myocardium, contributing to the overall health and function of the heart [124,125]. Additionally, miR-499 targets genes involved in the regulation of mitochondrial function and integrity. Proper mitochondrial function regulated by miR-499 ensures adequate energy production, essential for the high-energy demands of rat cardiomyocytes [126]. In vitro experiments have demonstrated that miRNA-499a-5p significantly improves DOX-induced CDTX via the CD38/MAPK/NF-κB signaling pathway [127]. miR-499 levels are elevated in sarcoma patients treated with DOX, who show decreased LVEF [128]. Furthermore, increases in miR-499, along with miR-1 and cytokines, have been demonstrated in DOX-treated in vivo murine models with decreased LVEF [128].

### 2.19. miR-9-5p

miR-9-5p has been recognized as a miRNA with potential roles in CDTX. It is involved in many cellular processes and pathways that can affect cardiac function and response to injury. miR-9-5p significantly influences the regulation of apoptosis, downregulating the *Bcl-2* anti-apoptotic gene [129]. Additionally, miR-9-5p plays a role in regulating cardiac fibrosis, targeting genes such as *COL1A1* and *TGF-β* and affecting the extent of fibrosis in the heart [130]. Its influence on fibrotic gene expression can lead to changes in the extracellular matrix and contribute to pathological cardiac remodeling. miR-9-5p also modulates inflammatory responses; this miRNA downregulates genes involved in inflammatory signaling pathways, such as *IL-10* [131], reducing the production of pro-inflammatory cytokines and decreasing inflammation. miR-9-5p can impact the extent of cardiac inflammation. Moreover, miR-9-5p is involved in regulating oxidative stress, targeting the *Nrf2* gene [132]. In addition, miR-9-5p can influence the cellular response to oxidative stress, reducing oxidative damage to mouse hearts [131]; this effect in oxidative stress could alleviate the effects of CDTX. In this sense, in vitro experiments have shown that miR-9-5p derived from human iPSC-MSCs exosomes decreases DOX-induced CDTX by inhibiting cardiomyocyte senescence [133].

### 2.20. miR-92a

miR-92a, a member of the miR-17-92 cluster, has been implicated in various cardiovascular pathologies and is considered a promising biomarker for CDTX. miR-92a plays a critical role in regulating angiogenesis, which is essential for cardiac repair and regeneration. miR-92a targets pro-angiogenic genes such as *ITGα5* and *eNOS* [134], leading to reduced endothelial cell migration, proliferation, and tube formation. Inhibiting these pro-angiogenic factors, miR-92a impairs vascular growth and repair processes, which are vital for maintaining cardiac function and healing following injury. miR-92a also affects endothelial function, which is crucial for vascular health and integrity, and downregulates *SIRT1* and *KLF2*, resulting in endothelial dysfunction and contributing to vascular inflammation and atherosclerosis [135]. The negative impact of miR-92a on endothelial cell function compromises the integrity of the endothelial barrier, making the cardiovascular system more susceptible to damage and CDTX. Additionally, miR-92a has been shown to promote inflammatory responses, upregulating inflammatory pathways by targeting anti-inflammatory genes such as *IL-10* [136] and *HO-1* [137], leading to increased production of pro-inflammatory cytokines and adhesion molecules. miR-92a exacerbates vascular injury and contributes to the development and progression of CDTX. miR-92a is also involved in regulating apoptosis in H9c2 cells, downregulating anti-apoptotic genes such as *MSK2* and *CREB3L2*, increasing the susceptibility of these cells to apoptosis [138]. The pro-apoptotic effects of miR-92a lead to increased cardiomyocyte death, contributing to the loss of functional cardiac tissue and worsening CDTX. miR-92a downregulates genes involved in the oxidative stress response, such as HO-1 [138], diminishing the antioxidant capacity of mouse and human endothelial cells. Downregulation of miR-92a-3p inhibits DOX-induced cardiac senescence by targeting *ATG4a*, a gene related to autophagy in human iPS-CM model in vitro [139].

## 3. Conclusions

The role of miRNAs as biomarkers for CDTX presents a promising avenue for the early detection and management of cardiovascular complications arising from various medical treatments, particularly cancer therapies. miRNAs offer several advantages as biomarkers, including their stability in body fluids, tissue-specific expression patterns, and ability to reflect pathological changes at the molecular level. The specific miRNAs (Table 1) discussed in this review—Let-7, miR-1, miR-106b-5p, miR-126, miR-129-5p, miR-133, miR-140, miR-143, miR-194-5p, miRNA-199a-5p, miR-200a, miR-208, miR-21, miR-210, miR-34a-5p, miRNA-4732-3p, miR-494-3p, miR-499, miR-9-5p, and miR-92a—highlight the diverse roles of these small RNA molecules in cardiovascular health and disease. There are many other miRNAs that have been previously identified to be related to CDTX, although they are not included in this review: miR-147, miR-204, miR-29, miR-675, miR-1303, miR-96, miR-130a, miR-181, miR-145, miR-324, miR-146, miR-624, miR-221, miR-409, miR-340, miR-376, miR-154, and miR-126. Among the miRNAs included in this review, it is worth highlighting, due to the greater number of studies focused on CVDs and regeneration, both miR-1 and miR-133, which have already identified as muscle-specific miRNAs crucial for cardiac function in both patients and animal models or in vitro cell line studies. Also noteworthy are the Let-7 family, widely studied in different animal models, as well as miR-140, miR-143, the miR-200 family, cardiac-specific miR-208, miR-21, and miR-34a-5p. The identification of specific miRNAs as biomarkers for CDTX has significant clinical implications. These miRNAs can aid in the early detection of cardiac injury, allowing for timely interventions to prevent or mitigate long-term cardiovascular complications. Furthermore, miRNAs have potential therapeutic applications, as modulating their expression could offer novel treatment strategies for CVDs.

To fully realize the potential of miRNAs as biomarkers and therapeutic targets, further research is needed to elucidate their precise mechanisms of action (Figure 3), validate their clinical utility, and develop reliable detection methods.

Large-scale clinical studies are essential to establish the sensitivity and specificity of miRNAs as biomarkers for CDTX and to identify the most effective therapeutic approaches for targeting these molecules, since most studies include small numbers of patients.

The use of miRNAs as biomarkers and/or therapeutic targets aligns with the principles of personalized medicine, where treatment strategies are tailored to the individual patient’s molecular profile. By integrating miRNA-based diagnostics and therapies into clinical practice, healthcare providers can offer more precise and effective management of CVDs, ultimately improving patient outcomes.

In conclusion, miRNAs represent a powerful tool in the fight against CDTX and CVDs. Their roles as regulators of gene expression, combined with their potential as biomarkers and therapeutic targets, underscore the importance of continued research and development in this field. As our understanding of miRNAs in cardiovascular health and disease deepens, we can expect to see significant advancements in the diagnosis, prognosis, and treatment of CDTX and related conditions.

## Figures and Tables

**Figure 1 ijms-25-11910-f001:**
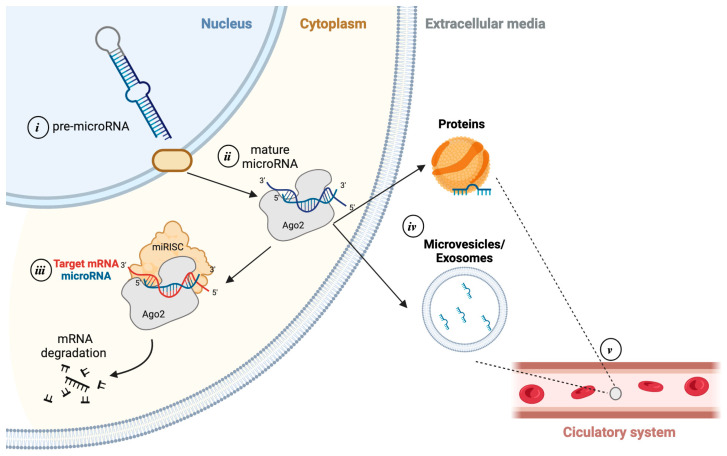
miRNAs biogenesis. (i) Biogenesis starts in the nucleus where the DNA is transcripted into pri-miRNAs, which are then cleaved into pre-miRNAs; (ii) pre-miRNAs are transported to the cytoplasm, where they are processed by the Argonaute complex into single-stranded mature miRNAs; (iii) mature miRNAs can be incorporated into the RNA-induced silencing complex (RISC), where they bind to target mRNAs to either inhibit translation or promote degradation; (iv) pre- and mature miRNAs can be secreted from the cell in exosomes and microvesicles or bound to RNA-binding proteins; (v) microvesicles containing miRNAs can be released to the circulatory system, allowing for its detection in blood and other body fluids.

**Figure 2 ijms-25-11910-f002:**
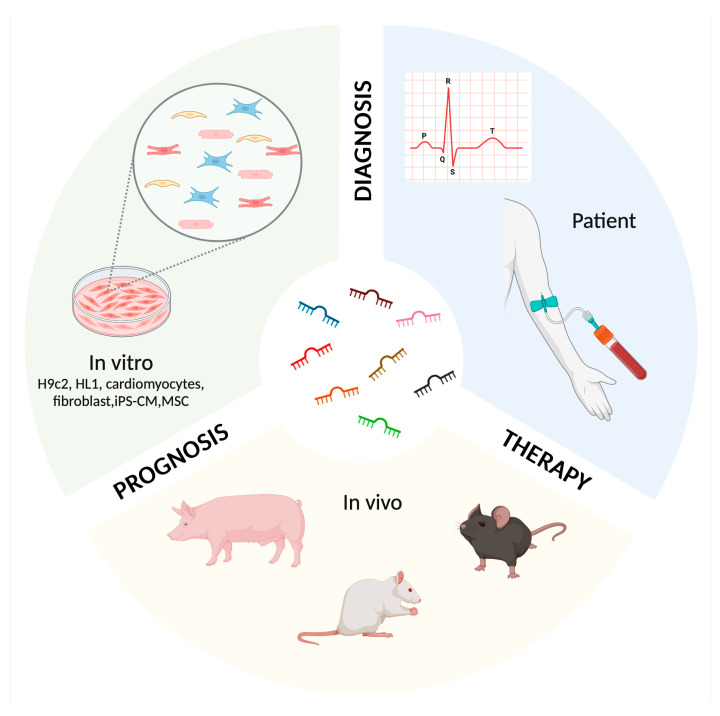
miRNA applications in patient, in vivo, and in vitro models.

**Figure 3 ijms-25-11910-f003:**
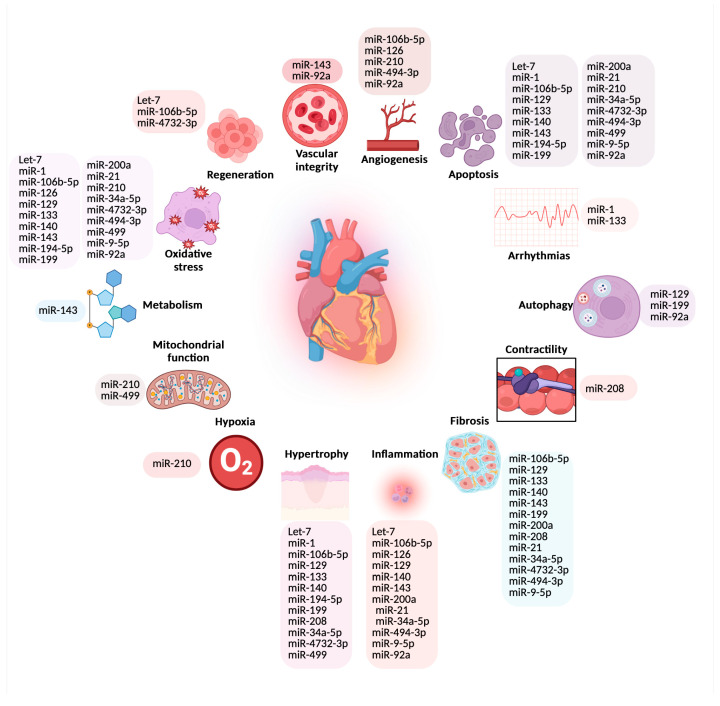
miRNAs and their role in cardiovascular cellular processes.

**Table 1 ijms-25-11910-t001:** miRNAs related to cancer therapy-induced CDTX.

MicroRNA	Species	Cell Type/Tissue	Treatment	Expression Change	Function	References
*Let-7*	Rat, pig, human	H9c2, cardiomyocytes, heart tissue, plasma	DOX	Decrease	Apoptosis, hypertrophy, inflammation, oxidative stress, regeneration	[19,20,21,22,23,24]
*miR-1*	Human, rat	Plasma, iPS-CM	DOX	Increase	Arrhythmias, hypertrophy, apoptosis, oxidative stress	[20,28,29]
*miR-106b-5p*	Mouse, human	iPS-CM, heart tissue	DOX	Increase	Apoptosis, regeneration, angiogenesis, fibrosis, oxidative stress, inflammation, hypertrophy	[33]
*miR-126*	Human	Patients, iPS-CM	DOX, neoadjuvant	Increase Decrease	Angiogenesis, inflammation, oxidative stress	[45,46,47,48]
*miR-129*	Rat, mouse	H9c2, HL-1	TKI, anthracyclines	Decrease	Apoptosis, autophagy, hypertrophy, fibrosis, inflammation, oxidative stress	[53,54]
*miR-133*	Mouse, human	HL-1, heart tissue, serum	Anthracyclines, DOX	Decrease	Hypertrophy, apoptosis, arrythmias, oxidative stress, fibrosis	[20,31,58]
*miR-140*	Rat, mouse, human	H9c2, serum, heart tissue, cardiomyocytes	DOX, bevacizumab	Increase	Apoptosis, hypertrophy, fibrosis, inflammation, oxidative stress	[20,64,65]
*miR-143*	Mouse, rat	H9c2, cardiomyocytes	DOX	Decrease	Apoptosis, vascular integrity, fibrosis, lipid and glucose metabolism, oxidative stress, inflammation	[74,75,76]
*miR-194-5p*	Mouse, rat	Heart tissue, H9c2	DOX	Decrease	Apoptosis, oxidative stress, hypertrophy	[79]
*miR-199*	Rat, human	Neonatal cardiomyocytes, plasma	Fluorouracil, neoadjuvant	Decrease Increase	Apoptosis, hypertrophy, autophagy, oxidative stress, fibrosis	[85,86]
*miR-200a*	Mouse, rats	Heart tissue, cardiomyocytes, H9c2	DOX	Decrease Increase	Apoptosis, oxidative stress, fibrosis, inflammation	[91,92,93]
*miR-208*	Rats, mouse	Heart tissue, plasma	DOX	Decrease Increase	Hypertrophy, fibrosis, contractility	[20,98]
*miR-21*	Mouse, rat, human	Heart tissue, serum, H9c2, cardiomyocytes, plasma	DOX	Decrease	Fibrosis, apoptosis, oxidative stress, inflammation	[20,101,102,103]
*miR-210*	Human	Patients, iPS-CM	Neoadjuvant	Decrease	Hypoxia, apoptosis, angiogenesis, mitochondrial function, oxidative stress	[24,47,48]
*miR-34a-5p*	Rats, mouse, human	H9c2, progenitor cells, aortic endothelial cells, heart tissue, plasma	DOX	Increase	Apoptosis, fibrosis, inflammation, hypertrophy, oxidative stress	[20,86,112,113]
*miR-4732-3p*	Human, rats	Serum, iPS-CM, cardiomyocytes	DOX	Decrease	Apoptosis, hypertrophy, fibrosis, oxidative stress, regeneration	[115,116]
*miR-494-3p*	Rat, mouse	H9c2, HL1	Pirarubicin	Increase	Apoptosis, fibrosis, inflammation, oxidative stress, angiogenesis	[121,122]
*miR-499*	Rat, mouse, human	H9c2, heart tissue, plasma	DOX	Decrease Increase	Apoptosis, hypertrophy, oxidative stress, mitochondrial function	[20,127,128]
*miR-9-5p*	Human	iPS, MSCs	DOX	Decrease	Apoptosis, fibrosis, inflammation, oxidative stress	[133]
*miR-92a*	Human	iPS-CM	DOX	Increase	Angiogenesis, vascular integrity, inflammation, apoptosis, oxidative stress, autophagy	[139]

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
