# Peer review of "Transforming Cardiotoxicity Detection in Cancer Therapies: The Promise of MicroRNAs as Precision Biomarkers"

_ijms, 2024, doi:10.3390/ijms252211910_

Round 1
Reviewer 1 Report
Comments and Suggestions for Authors
The review needs strong revision.
1. First of all, the origin of the miRNA being discussed is unclear. Are the authors referring only to human tissues and cells? The title suggests the use of 'precision biomarkers' likely for diagnosing human cardiotoxicity, not for rat or mouse models, which implies that only human samples should be considered. If this is not the case, it should be explicitly stated whether the miRNA was identified in human or laboratory animal tissues or cells. Currently, this distinction is not clear, and the information seems to be mixed. For instance, the term 'cardiac cells' is used frequently, but it is unclear which specific cell types in the heart the authors are referring to, given the various types of cells present. This is just one example of many vague or imprecise statements throughout the text. Additionally, it should be clearly stated whether the studies were conducted in vitro or in vivo, and whether they involved tissues or cells. The type of models used should also be specified to avoid ambiguity.
2. Including a figure that illustrates the functioning of miRNAs would also enhance this review.
3. The review would also benefit from a table summarizing the data, including key information such as the type of miRNA, its origin, the model used, and its main functions. This would provide a clearer and more organized overview for the reader.
4. The conclusions are superficial and lack specific points regarding which miRNAs are most relevant and for which specific cardiac functions or dysfunctions. More concrete recommendations and insights into the best miRNAs for various cardiac processes should be provided.
Author Response
Response to Reviewer 1 Comments
- Summary
We sincerely thank the reviewer for constructive criticisms and valuable comments which were of great help in revising the manuscript. We have rewritten and modified the manuscript as reviewer indicated, including figures and a table. We consider that manuscript has improved and all points were addressed according reviewer’ suggestions as it is described below.
|
2. Questions for General Evaluation |
Reviewer’s Evaluation |
Response and Revisions |
|
Is the work a significant contribution to the field? |
|
|
|
Is the work well organized and comprehensively described? |
|
|
|
Is the work scientifically sound and not misleading? |
|
|
|
Are there appropriate and adequate references to related and previous work? |
|
|
|
Is the English used correct and readable? |
|
|
- Point-by-point response to Comments and Suggestions for Authors
Comments 1: First of all, the origin of the miRNA being discussed is unclear. Are the authors referring only to human tissues and cells? The title suggests the use of 'precision biomarkers' likely for diagnosing human cardiotoxicity, not for rat or mouse models, which implies that only human samples should be considered. If this is not the case, it should be explicitly stated whether the miRNA was identified in human or laboratory animal tissues or cells. Currently, this distinction is not clear, and the information seems to be mixed. For instance, the term 'cardiac cells' is used frequently, but it is unclear which specific cell types in the heart the authors are referring to, given the various types of cells present. This is just one example of many vague or imprecise statements throughout the text. Additionally, it should be clearly stated whether the studies were conducted in vitro or in vivo, and whether they involved tissues or cells. The type of models used should also be specified to avoid ambiguity.
Response 1: Thank you for pointing this out, we totally agree with reviewer´s comment. Therefore, we have changed throughout the text and specified the in vitro and in vivo models used in each study, if it has been used tissue or cells, with kind of cardiac cell or if it is a basic study or patient study, also we have included a table (Table 1) in which species and kind of tissue or cell has been used was specified. The title is referring to possible microRNAs that could be used as precision biomarkers, based on mechanistic pathways or functions in which these microRNAs are implicated that may be involved in the development of cardiotoxicity of cancer treatments. Also, each microRNA paragraph includes several studies that demonstrates their implication in cardiotoxicity, but in cell culture, in vivo experiments or circulating in patients. We have modified the text in order to clarify all the aspects indicated by the reviewer and also included 3 figures and a table in order to illustrate these models.
Comments 2: Including a figure that illustrates the functioning of miRNAs would also enhance this review.
Response 2: Accordingly with the reviewer, we have included 3 different figures throughout the text to illustrate the manuscript, and also included a graphical abstract. Figure 1 about miRNAs biogenesis, it has been included in page 2 line 62. Also, we have included Figure 2 about miRNA applications in patient, in vivo and in vitro models (page 3 line 106). To finish, we also included figure 3 about microRNAs and their role in cardiovascular cellular processes, in page 17 line 779.
In addition, a paragraph about microRNAs biogenesis has been included in Introduction section, page 2, line 51:
“The biogenesis of miRNAs starts with the transcription of pri-miRNA by RNA polymerase II or III. This long transcript (~100 bp) is processed in the nucleus by the enzyme DROSHA into pre-miRNA (~70 bp). Pre-miRNAs are exported to the cytoplasm via exportin 5, where they are processed by the RNase enzyme DICER, which breaks the terminal loop, producing a mature RNA duplex (~22 bp). The less stable strand is loaded into Argonaute proteins (AGO2), forming the RNA-induced silencing complex (RISC). This complex enables the miRNA to bind to target mRNAs, leading to their degradation or inhibition of translation, depending on the degree of base pairing. A single miRNA can regulate thousands of mRNAs, and conversely, an mRNA can be targeted by many miRNAs (Figure 1) [7].”
Comments 3: The review would also benefit from a table summarizing the data, including key information such as the type of miRNA, its origin, the model used, and its main functions. This would provide a clearer and more organized overview for the reader.
Response 3: Agreeing with the reviewer we have also included a table to summarize all microRNAs, including key information such as species, cell type/tissue, main functions, treatment, expression change, and references (page 14, line 760).
Comments 4: The conclusions are superficial and lack specific points regarding which miRNAs are most relevant and for which specific cardiac functions or dysfunctions. More concrete recommendations and insights into the best miRNAs for various cardiac processes should be provided.
Response 4: According to reviewer’s comment, we have modified the conclusions of the paper.
Including a paragraph with few more microRNAs which are not listed in the article, but that were previously related with cardiotoxicity, as reviewer 2 suggested (page 14, line 743).
“There are many other miRNAs that have been previously identified to be related to CDTX, although not included in this review: miR-147, miR-204, miR-29, miR-675, miR-1303, miR-96, miR-130a, miR-181, miR-145, miR-324, miR-146, miR-624, miR-221, miR-409, miR-340, miR-376, miR-154 or miR-126.”
In addition, figure 3 and a table have been included in conclusions section, as previously indicated.
As reviewer suggested, we have also included a paragraph specifying the most relevant microRNAs (page 14, line 747) “Among the microRNAs included in this review, it is worth highlighting, due to the greater number of studies focused on cardiovascular diseases and regeneration, both miR-1 and miR-133, already identified as muscle-specific miRNA and crucial for cardiac function; both in patients and in animal models or in vitro cell lines studies. Also note-worthy are the Let-7 family, widely studied in different animal models, as well as miR-140, miR-143, the miR-200 family, cardiac specific miR-208, miR-21 and miR-34a-5p.”
- Response to Comments on the Quality of English Language
While reviewing the entire manuscript, some parts of the text have been modified to improve English grammar and comprehension.

Reviewer 2 Report
Comments and Suggestions for Authors
Reviewer report
Transforming Cardiotoxicity Detection: The Promise of Mi-croRNAs as Precision Biomarkers
This is a well-organized study showing the importance of miRNAs represent a powerful tool in the fight against cardiotoxicity in clinical and pre-clinical areas. Article focuses lights on the role of miRNAs as biomarkers for cardiotoxicity and explores specific miRNAs implicated in cardiovascular health and disease, their mechanisms of action. Articles needs to be improved in some respects.
A. This article lacks figures please describe few diagrams and figures related to mircro RNA depicting their role in the cardiotoxicity detection.
B. Please describe the entire article in the form of graphical abstract.
C. Please describe respective all the micro-RNA mentioned in article in the form of table.
D. How A systematic literature review was carried out please describe in detail with the databases followed.
E. Are there few more micro RNAs which are not listed in the article please list out briefly.
F. Please follow the below mentioned article to improve the current manuscript in the aspects of figure, table.
Brown C, Mantzaris M, Nicolaou E, Karanasiou G, Papageorgiou E, Curigliano G, Cardinale D, Filippatos G, Memos N, Naka KK, Papakostantinou A. A systematic review of miRNAs as biomarkers for chemotherapy-induced cardiotoxicity in breast cancer patients reveals potentially clinically informative panels as well as key challenges in miRNA research. Cardio-Oncology. 2022 Sep 7;8(1):16.
Author Response
Response to Reviewer 2 Comments
- Summary
Thank you very much for taking the time to review this manuscript. Please find the detailed responses below and the corresponding revisions/corrections highlighted/in track changes in the re-submitted files.
|
2. Questions for General Evaluation |
Reviewer’s Evaluation |
Response and Revisions |
|
Is the work a significant contribution to the field? |
|
|
|
Is the work well organized and comprehensively described? |
|
|
|
Is the work scientifically sound and not misleading? |
|
|
|
Are there appropriate and adequate references to related and previous work? |
|
|
|
Is the English used correct and readable? |
|
|
- Point-by-point response to Comments and Suggestions for Authors
Comments 1: This article lacks figures please describe few diagrams and figures related to microRNA depicting their role in the cardiotoxicity detection.
Response 1: We completely agree with reviewers´ comment. Therefore, I have included 3 different figures to illustrate the role of microRNAs in cardiotoxicity and also a table to summarized the data of all microRNAs included in this review paper.
Figure 1 about miRNAs biogenesis, it has been included in page 2 line 62. Also, we have included Figure 2 about miRNA applications in patient, in vivo and in vitro models (page 3 line 106). To finish, we also included figure 3 about microRNAs and their role in cardiovascular cellular processes, in page 17 line 779. We have also included a table to summarize all microRNAs, including key information such as species, cell type/tissue, main functions, treatment, model, expression change, and references (page 14, line 760).
Comments 2: Please describe the entire article in the form of graphical abstract.
Response 2: Thank you for this comment. We have included a graphical abstract as reviewer suggested.
Comments 3: Please describe respective all the microRNA mentioned in article in the form of table.
Response 3: We appreciate the reviewer's comments. As indicated in response 1 we have included a table to summarized the data of all microRNAs included in this review paper (page 14, line 760).
Comments 4: How A systematic literature review was carried out please describe in detail with the databases followed.
Response 4: We appreciate the reviewer’s concern. This is a literature review, not a systematic review, that why we did not indicate the databases and terms of the search. We first performed a search in PubMed, Google Scholar, Web of Science (WoS) or Scopus using the terms microRNAs and cardiotoxicity. Then we excluded papers that were not related to cancer therapies. After, we chose the microRNAs that are included in the paper based in the frequency that appear in the literature, we included the 20 more studied microRNAs. We followed the recommendations of other studies in which it is shown how to perform a scoping review, consisting in 5 stages, Stage 1: identifying the research question; Stage 2: identifying relevant studies; Stage 3: study selection; Stage 4: charting the data; Stage 5: collating, summarizing and reporting the results (Arksey, H., & O’Malley, L. (2005). Scoping studies: towards a methodological framework. International Journal of Social Research Methodology, 8(1), 19–32. https://doi.org/10.1080/1364557032000119616).
Comments 5: Are there few more microRNAs which are not listed in the article please list out briefly.
Response 5: According to reviewer’s comment, we modified the text to include this point. This change can be found in page number 14, and line 743.
“There are many other miRNAs that have been previously identified to be related to CDTX, although not included in this review: miR-147, miR-204, miR-29, miR-675, miR-1303, miR-96, miR-130a, miR-181, miR-145, miR-324, miR-146, miR-624, miR-221, miR-409, miR-340, miR-376, miR-154 or miR-126.”
Comments 6: Please follow the below mentioned article to improve the current manuscript in the aspects of figure, table.
Brown C, Mantzaris M, Nicolaou E, Karanasiou G, Papageorgiou E, Curigliano G, Cardinale D, Filippatos G, Memos N, Naka KK, Papakostantinou A. A systematic review of miRNAs as biomarkers for chemotherapy-induced cardiotoxicity in breast cancer patients reveals potentially clinically informative panels as well as key challenges in miRNA research. Cardio-Oncology. 2022 Sep 7;8(1):16.
Response 6: Accordingly, we have included a graphical abstract, 3 figures and a table using this paper as a model to improve the manuscript.
- Response to Comments on the Quality of English Language
While reviewing the entire manuscript, some parts of the text have been modified to improve English grammar and comprehension.

Reviewer 3 Report
Comments and Suggestions for Authors
The authors presented a narrative review about the role of microRNAs as precision biomarkers for cardiotoxicity detention. Cariotoxicity is significantly influenced by microRNAs, especially when it comes to hazardous exposures and cancer treatments. They are prospective biomarkers and therapeutic targets since their dysregulation might have detrimental cardiac effects.
Could authors specify search terms and databases used to search literature? Nothing about method is illustrated.
I recommend to authors to consider the role of antidepressants in this mechanism: most antidepressants have negative effects on the cardiovascular system. They may have an impact on cardiac conduction, blood pressure, and the heart's inotropic condition. These agents' direct activities or pharmacological interactions may result in adverse responses (10.1016/0028-3908(80)90201-4). It is very important to consider, especially in older adults (10.1186/s12877-020-01730-5).
Author Response
Response to Reviewer 3 Comments
- Summary
We want to thank reviewer for his/her exhaustive comments. We realize that there are some aspects that we have not clarified enough. We have edited the manuscript considering his/her commentaries. We believe that his/her point of view has improved the message. We hope that the reviewer will give us a new opportunity.
|
2. Questions for General Evaluation |
Reviewer’s Evaluation |
Response and Revisions |
|
Is the work a significant contribution to the field? |
|
|
|
Is the work well organized and comprehensively described? |
|
|
|
Is the work scientifically sound and not misleading? |
|
|
|
Are there appropriate and adequate references to related and previous work? |
|
|
|
Is the English used correct and readable? |
|
|
- Point-by-point response to Comments and Suggestions for Authors
Comments 1: Could authors specify search terms and databases used to search literature? Nothing about method is illustrated.
Response 1: We appreciate the comment, we first performed a search in PubMed, Google Scholar, Web of Science (WoS) or Scopus using the terms microRNAs and cardiotoxicity. Then we excluded papers that were not related to cancer therapies. After, we chose the microRNAs that are included in the paper based in the frequency that appear in the literature, we included the 20 more studied microRNAs, there are many others that could be included, but a list of other possible microRNAs that could be used as biomarkers is now included in the text (page 14 line 743). Following the recommendations of other studies in which it is shown how to perform a scoping review, consisting in 5 stages, Stage 1: identifying the research question; Stage 2: identifying relevant studies; Stage 3: study selection; Stage 4: charting the data; Stage 5: collating, summarizing and reporting the results (Arksey, H., & O’Malley, L. (2005). Scoping studies: towards a methodological framework. International Journal of Social Research Methodology, 8(1), 19–32. https://doi.org/10.1080/1364557032000119616).
Comments 2: I recommend to authors to consider the role of antidepressants in this mechanism: most antidepressants have negative effects on the cardiovascular system. They may have an impact on cardiac conduction, blood pressure, and the heart's inotropic condition. These agents' direct activities or pharmacological interactions may result in adverse responses (10.1016/0028-3908(80)90201-4). It is very important to consider, especially in older adults (10.1186/s12877-020-01730-5).
Response 2: Thank you for pointing this out. This review is focused only in cardiotoxicity effects of cancer therapies, that is why the title has been modified in order to clarify this aspect including in cancer treatments. Although the issue suggested by the reviewer is very interesting and also demonstrate the effects of antidepressant in cardiovascular health, mainly in elderly patients, we found a low number of papers relating microRNAs and this kind of treatments, we have already included this reference in the manuscript, relating mental disorder treatments and microRNA expression:
“Abdel Hamid OI, Ibrahim EM, Hussien MH, ElKhateeb SA. The molecular mechanisms of lithium-induced cardiotoxicity in male rats and its amelioration by N-acetyl cysteine. Hum Exp Toxicol. 2020 May;39(5):696-711. doi: 10.1177/0960327119897759.”
We have included a sentence regarding this issue proposed by reviewer: “In this sense, together with miR-1 and miR-29, miR-21 is increased in rat in vivo experiments treated other cardiotoxic medications such as antidepressants that also increase oxidative stress [101]” (page 10 line 525).
- Response to Comments on the Quality of English Language
While reviewing the entire manuscript, some parts of the text have been modified to improve English grammar and comprehension.
- Additional clarifications
We would like to point out that we have also included 3 summary figures to illustrate the implication of miRNAs in cardiotoxicity and a table summarizing the data of each microRNA. Also, a graphical abstract has been included.

Round 2
Reviewer 1 Report
Comments and Suggestions for Authors
The review has been significantly improved, however, some corrections still are required mainly related to the "cardiac cells". Remarks:
2.3. miR-106b-5p section, as well as all other, are still mentioning “cardiac cells”. If the authors are talking about all types of the cardiac cells that means they are taking about the cardiac tissue, not about the individual cells since cardiac tissue, is composed of various types of cells including cardiomyocytes, fibroblasts/mesenchymal stem/stromal cells, endothelial, pericytes, mesangyoblasts and other. Writing “cardiac cells” is not clear about what type of the cells or tissue and what origin (human, rat, mice or other) the authors are tanking about. The citation 33 is about cancer cells.
Section 2.4 is also mentioning “cardiac cells”, while citation 43 is about endothelial cells.
Section 2.5 – “cardiac cells” are human cardiomyocytes (citation 49).
Section 2.8. – „cardiac cells“, citation 65, is about rat neonatal cardiomyocytes. Why not to clearly write that instead of general „cardiac cells“ ? Citation 70 is about preadipocytes.
Section 2.10 – ‚cardiac cells‘ are cardiomyocytes (the origin should be checked).
Section 2.14 – „cardiac cells“ are cancer cells (citation 104).
Section 2.15 – „cardiac cells“, citation 108 is animal model.
Section 2.16 – „cardiac cells“, citation 111 are cholangiopathies and MSC EVs.
Section 2.17 - „cardiac cells“ , citation 116, are rat neonatal cardiomyocyte culture.
Section.....and so on. The authors should carefully check what they are citing and writing.
Author Response
Response to Reviewer 1 Comments. R2
- Summary
We want to thank the Reviewer 1 for his/her exhaustive comments. We want to apologize to the reviewer for not having clearly reviewed some parts of the manuscript. We realize that there were some aspects that we have not clarified enough. We have modified bellow following specific commentaries.
|
2. Questions for General Evaluation |
Reviewer’s Evaluation |
Response and Revisions |
|
Is the work a significant contribution to the field? |
|
|
|
Is the work well organized and comprehensively described? |
|
|
|
Is the work scientifically sound and not misleading? |
|
|
|
Are there appropriate and adequate references to related and previous work? |
|
|
|
Is the English used correct and readable? |
|
|
- Point-by-point response to Comments and Suggestions for Authors
Comments 1: The review has been significantly improved; however, some corrections still are required mainly related to the "cardiac cells". Remarks:
2.3. miR-106b-5p section, as well as all other, are still mentioning “cardiac cells”. If the authors are talking about all types of the cardiac cells that means they are taking about the cardiac tissue, not about the individual cells since cardiac tissue, is composed of various types of cells including cardiomyocytes, fibroblasts/mesenchymal stem/stromal cells, endothelial, pericytes, mesangyoblasts and other. Writing “cardiac cells” is not clear about what type of the cells or tissue and what origin (human, rat, mice or other) the authors are talking about. The citation 33 is about cancer cells.
Section 2.4 is also mentioning “cardiac cells”, while citation 43 is about endothelial cells.
Section 2.5 – “cardiac cells” are human cardiomyocytes (citation 49).
Section 2.8. – “cardiac cells”, citation 65, is about rat neonatal cardiomyocytes. Why not to clearly write that instead of general “cardiac cells”? Citation 70 is about preadipocytes.
Section 2.10 – “cardiac cells” are cardiomyocytes (the origin should be checked).
Section 2.14 – “cardiac cells” are cancer cells (citation 104).
Section 2.15 – “cardiac cells”, citation 108 is animal model.
Section 2.16 – “cardiac cells”, citation 111 are cholangiopathies and MSC EVs.
Section 2.17 – “cardiac cells”, citation 116, are rat neonatal cardiomyocyte culture.
Section....and so on. The authors should carefully check what they are citing and writing.
Response 1: According to reviewer’s comments, we have modified and reorganized the manuscript, including references and some parts throughout the text.
2.3. miR-106b-5p section: the term “cardiac cells” was changed to H9c2, animal models and human heart, cancer cells and neonatal mouse cardiomyocytes in each case.
Section 2.4 is also mentioning “cardiac cells”, while citation 43 is about human endothelial cells.
Section 2.5 – “cardiac cells” are human cardiomyocytes (citation 49).
Section 2.8. – “cardiac cells”, citation 65, is about rat neonatal cardiomyocytes. Why not to clearly write that instead of general “cardiac cells”? Citation 70 is about mouse preadipocytes. Also, smooth muscle cells.
Section 2.10 – “cardiac cells” are rat cardiomyocytes (the origin should be checked).
Section 2.14 – “cardiac cells” are cancer cells (citation 104).
Section 2.15 – “cardiac cells”, citation 108 is animal model. The terms were changed to H9c2 cells and rat CDTX-model, human smooth muscle cells.
Section 2.16 – “cardiac cells”, citation 111 are cholangiopathies and MSC EVs. The terms were changed to from extracellular vesicles derived from human mesenchymal stromal cells, rat heart injury models and neonatal rat cardiomyocytes
Section 2.17 – “cardiac cells”, citation 116, are rat neonatal cardiomyocyte culture.
All these sections have been modified according to reviewer comments, also many other sections were also modified in order to change the term “cardiac cell” into a specific cell type/tissue and origin. Also, other sections and references were modified or included, as indicated:
Section 2.2 – “cardiac cells” are cardiomyocytes, rat heart tissue and H9c2.
Section 2.6 – “cardiac cells” are rat cardiomyocytes and H9c2
Section 2.7 – “cardiac cells” are mouse heart
Section 2.9 – “cardiac cells” are H9c2 cells
Section 2.13 – “cardiac cells” are endothelial cells
Section 2.14 – “cardiac cells” references including H9c2 cells were added
Section 2.15 – “cardiac cells” are H9c2 cells and human smooth muscle cells
Section 2.18 – “cardiac cells” are plasma samples from patients, neonatal rat cardiomyocytes, human hearts and rat cardiomyocytes
Section 2.19 – “cardiac cells” are mice hearts
Section 2.20 – “cardiac cells” are H9c2 cells and mouse and human endothelial cells